# Prediction of Lumbar Disc Bulging and Protrusion by Anthropometric Factors and Disc Morphology

**DOI:** 10.3390/ijerph18052521

**Published:** 2021-03-04

**Authors:** Isabella Yu-Ju Hung, Tiffany Ting-Fang Shih, Bang-Bin Chen, Yue Leon Guo

**Affiliations:** 1Department of Nursing, Chung Hwa University of Medical Technology, Tainan 71703, Taiwan; isabella@ms.hwai.edu.tw; 2Department of Medical Imaging and Radiology, National Taiwan University (NTU) Hospital and NTU College of Medicine, Taipei 100225, Taiwan; ttfshih@ntu.edu.tw (T.T.-F.S.); bangbinchen@ntu.edu.tw (B.-B.C.); 3National Institute of Environmental Health Sciences, National Health Research Institute (NHRI), Miaoli 35053, Taiwan; 4Department of Environmental and Occupational Medicine, College of Medicine, National Taiwan University (NTU) and NTU Hospital, Taipei 100225, Taiwan; 5Graduate Institute of Environmental and Occupational Health Sciences, College of Public Health, National Taiwan University, Taipei 100225, Taiwan

**Keywords:** predictive model, disc bulging, disc protrusion, morphology, receiver operating characteristic (ROC) curve, anthropometric factors

## Abstract

The relationship between reduced disc height and disc bulging and/or protrusion has been controversial. The purposes of this study were to examine the relationship between disc morphology and disc bulging and protrusion and to establish a model for predicting disc bulging and protrusion. This is a retrospective study. A total of 452 MRI scans from a spine study were analysed, 210 (46.5%) were men. Logistic regression analysis was applied to identify the association between anthropometric factors, disc morphology factors, and outcome. Model 1 was constructed using anthropometric variables to investigate the capacity for predicting outcomes. Model 2 was constructed using anthropometric and disc morphology variables. Age, body weight, body height, disc height, and disc depth were significantly associated with outcome. The area under the curve (AUC) statistics of Model 2 were significantly better than those of Model 1 at the L3-L4 and L4-L5 levels but not at the L5-S1 level. The results showed an association between disc morphology and disc bulging and/or protrusion at the L3-L4, L4-L5, and L5-S1 levels. The model utilizing both anthropometric factors and disc morphology factors had a better capacity to predict disc bulging and/or protrusion compared with the model using only anthropometric factors.

## 1. Introduction

Lumbar disc degeneration (LDD) is among the most important conditions causing low back pain (LBP), and the medical cost associated with LBP is enormous in developed countries [1,2,3,4,5]. Among disc degeneration conditions, herniated nucleus pulpolus or herniated intervertebral disc (HIVD) is one of the most commonly diagnosed abnormalities associated with LBP and sciatica [6]. In the clinical setting, the radiographic diagnosis of HIVD usually requires magnetic resonance imaging (MRI), which is less readily available in most primary care facilities. Although plain X-rays do not provide information on disc conditions and only a reduction in disc height is visible [7], X-rays are still a relatively inexpensive way to confirm suspicion of a herniated disc. If a suspicion is strengthened, other methods may be used to provide final confirmation.

Since X-rays are frequently obtained on patients with back pain, if a relationship between reduced disc height and disc bulging and/or protrusion could be established, useful insight could be obtained to guide further directions for patient evaluation. However, this relationship has been controversial; some studies found an association between disc height reduction and disc bulging [8,9] or herniation [10], but other studies did not observe such an association [11,12]. In Videman’s study, disc herniation was associated with larger axial disc area [13]. Regarding anthropometric factors, Hrubec reported that body height and body weight were positively associated with the risk of disc herniation [14]. Other studies also indicated that tallness is a factor associated with increased risk of herniation [14,15]. In addition, a number of studies have indicated that ageing is an important risk factor for disc bulging and/or protrusion [12,14,16,17,18]. From the literature review, disc morphology such as disc height and disc depth, and the anthropometric factors like age, sex, body height, and body weight had the relationship been LDD and disc herniation.

The purpose of this study was to examine whether disc morphology could provide useful information for the prediction of disc bulging and/or protrusion while controlling for anthropometric factors such as age, sex, body height, and body weight, which have been associated with LDD and disc herniation [14,15,16,17,18]. Furthermore, if such a relationship was present, this study also aimed to establish a prediction model using anthropometric factors and disc morphology to predict disc bulging and/or protrusion.

## 2. Materials and Methods

### 2.1. Subjects Recruitment

This study is a part of a project designed to investigate spine and bone disorders. Before participating in the study, the subjects received written and oral information regarding the study procedures and potential adverse effects and signed published informed consent forms. The study protocols and consent forms were reviewed and approved by the National Taiwan University Hospital Research Ethics Committee (NTUH-REC No.:200805047R).

We performed the sample size calculation in the study design phase. We planned to recruit 568 participants for this study based on a sample size estimation with a power of 80% and an alpha level of 0.05 to detect an odds ratio (OR) of 1.3 for patients with disc bulging and/or protrusion. The subjects were recruited from among wholesale market workers and walk-in clinic patients who sought treatment in the Internal Medicine clinic and were diagnosed with upper respiratory infections (URIs), mostly the common cold. During recruitment, the volunteers were not informed of the hypothesis of the study. They were invited to participate in a survey regarding spine and bone health from 2009 to 2011. The inclusion criteria of the project were that the subjects were between 20 and 65 years of age and had at least six months of work experience. A subject was excluded if he or she had previously been diagnosed with cancer, psychiatric conditions, spinal tumours, inflammatory spondylopathy, compression fractures, or major back trauma. Further, those with degenerative spondylolisthesis, spondylolytic spondylolisthesis and/or a previous lumbar spine operation were excluded from this study due to potential inaccuracy in the measurements of disc height, disc depth, and the anteroposterior diameter of the intervertebral disc, which may change due to the sliding of vertebral bodies or an operation. Of the 553 participants with MRI scans from the spine and bone study, 101 had spondylolisthesis and/or a previous lumbar spine operation at any lumbar level and were excluded from the analysis, leaving 452 participants with MRI scans for statistical analysis in this study.

### 2.2. Data Collection

Every participant received an assessment with a questionnaire and MRI of the lumbar spine. The content of the questionnaire each participant completed consisted of questions regarding demographic information including age, sex, body height, and body weight, history of back pain, driving and riding hours per week, exercise activities, drinking and smoking histories, occupational data including job titles, working tenures, descriptions of tasks, lifting exposure at work (e.g., estimates of the most common weights lifted or carried), frequency and duration of lifting or carrying, number of working hours per day, and number of working days per week.

#### 2.2.1. Magnetic Resonance Imaging (MRI) Equipment and Protocol

All MRI examinations were obtained at National Taiwan University Hospital (NTUH) with a GE 1.5-T unit (General Electric Medical Systems, Milwaukee, WI, USA) and using a spine array coil (5 × 11 in.) during 2009–2011. The studies consisted of four spin-echo sequences: an axial localizer (spoiled gradient), sagittal views with a repetition time and echo time (TR/TE) of 500/minimum full msec and 3350/110 msec, and an axial view with a TR/TE of 5325/110 msec. Technical specifications included a slice thickness of 4 mm for sagittal and axial sequences; a field of view (FOV) of 28 and 20 cm for the sagittal and axial images. The first thoracic vertebra, T1-weighted axial sequences were stacked slices extending from the inferior aspect of the twelfth thoracic vertebra (T12) through the inferior aspect of the first sacral spine (S1). There were two excitations for the T1-weighted axial and sagittal images, with one excitation for the second thoracic vertebra, T2-weighted sagittal images.

#### 2.2.2. Definition of Disc Bulging, Disc Protrusion, Degenerative Spondylolisthesis and Spondylolytic Spondylolisthesis

Intervertebral discs at the L3-L4, L4-L5, and L5-S1 levels (the third, fourth and fifth lumbar spine vertebrae and the first sacral vertebra) were evaluated for disc bulging and protrusion. These levels were investigated in this study as degeneration occurs most often and earliest in these three lower vertebral levels [17,19]. The outcome in this study was set as disc bulging and/or protrusion. An experienced radiologist performed the evaluation based on standard images according to written instructions. The radiologist was blinded to the participants’ medical histories and occupational exposure statuses. Disc bulging was defined as the presence of disc tissue that is circumferentially (50–100%) beyond the edges of the ring apophyses. Protrusion was defined as when the greatest distance, in any plane, between the edges of the disc material beyond the disc space was more than the distance between the edges of the base in the same plane [20]. The intrarater reliability regarding the presence or absence of disc bulging and disc protrusion was determined using 60 participants evaluated on 2 occasions within 3 months. Degenerative spondylolisthesis was defined as displacement of one vertebral body relative to the next inferior vertebral body with an intact pars interarticularis, and spondylolytic spondylolisthesis was defined as the separation of the posterior aspect of the vertebral body from the anterior body [21].

#### 2.2.3. Disc Morphology Measurement

There have been numerous reports of measurements of the lumbar disc height and disc depth [10,22,23,24,25]. Based on previous studies, we calculated disc height as the mean of the anterior and posterior disc heights and disc depth as the mean of the superior and inferior disc depth. The disc morphology measurements were based on bony structures, not the observed disc. This allows for development of prediction methods using plain films in the future. For the L3-L4 disc, L4-L5 disc, and L5-S1 disc, the measurements were carried out by two investigators and are illustrated in Figure 1 as follows:

(1) Anterior disc height (ADH) was measured from the anterior corners of the adjacent superior and inferior vertebral bodies.

(2) Posterior disc height (PDH) was measured from the posterior corners of the adjacent superior and inferior vertebral bodies.

(3) Disc height (DH) was calculated as the mean of the ADH and PDH, using the method described by Dabbs [23].

(4) Superior disc depth (SDD) was measured as the inferior distance between the anterior and posterior corners of the upper vertebral body.

(5) Inferior disc depth (IDD) was measured as the superior distance between the anterior and posterior corners of the lower vertebral body.

(6) Disc depth (DD), or the anteroposterior diameter of the intervertebral disc, was taken as the mean of the SDD and IDD [10,22,25].

The lines providing the DD and DH measures were drawn by the computer based on a standard algorithm. DD and DH measurements were performed three times in a subgroup of 57 subjects by two readers. One reader performed two measurements at 14-day intervals, allowing for the calculation of intrarater reliability. The results of the second reader were compared with the mean result of the DD and DH measurements of the first reader so interrater reliability could be ascertained.

### 2.3. Statistical Analysis

All statistical analyses were performed with the statistical software JMP version 5.0 (SAS Institute Inc, Cary, NC, USA). Categorical variables are expressed as frequencies and percentages, and continuous variables are expressed as means and range. Percentage agreement was used to evaluate MRI reproducibility of disc bulging and/or protrusion. The reliabilities of the digitizing procedures within examiners at a 2-week interval and between examiners were assessed with the intraclass correlation coefficient (ICC). Logistic regression analysis was applied to identify the association between anthropometric factors, disc morphology factors and disc bulging and/or protrusion at the L3-L4, L4-L5, and L5-S1 levels. For model development, a univariate analysis was used for the initial variable selection, and all potential variables with *p* < 0.25 were further evaluated using multivariable logistic regression. The potential variables included age, sex, body height, body weight, tenure, educational level, lifting exposure, exercise, smoking, drinking, and driving. We used a backward stepwise approach for the multivariable logistic regression. Age was highly related to tenure; thus, we excluded tenure as a factor to prevent overestimation. In addition, variables that were based on the findings from past research were also selected for use in the model, such as body height, were reported that associated with disc herniation [14]. Model 1 (Anthropometric model) was built with anthropometric variables including age, sex, body height and body weight to investigate the capacity for predicting disc bulging and/or protrusion. Model 2 (Disc morphology and anthropometric model) was built with anthropometric variables and disc morphology variables, including DH and DD, for predicting disc bulging and/or protrusion. *p* < 0.05 was considered to be statistically significant. The ability of the models to discriminate between participants with and without disc bulging and/or protrusion was evaluated by receiver operating characteristic (ROC) curve analysis. Models with area under the curve (AUC) statistics equal to 0.5 were not considered better than chance alone, whereas models with higher AUC statistics were considered better than chance. We then compared the AUC for different models with the Wilcoxon-Mann-Whitney U test, using MedCalc for Windows version 9.2.1.0 (MedCalc Software, Mariakerke, Belgium).

## 3. Results

Of the 452 participants with MRI scans in this study, 210 (46.5%) were men and 242 (53.5%) were women. The mean age was 49.3 years (standard deviation [SD]: 10.5). Their demographic data were shown in Table 1. The average tenure was 22.5 (0.63–50) years. The prevalence rates of disc bulging and/or protrusion at L3-L4, L4-L5, and L5-S1 were 44.0%, 60.4%, and 43.6%, respectively. None of the participants were found to have extrusion or sequestration. There were 313 (70.0%) subjects experiencing low back pain within the past 12 months in this study. The disc morphology factors are shown in Table 2. The mean DHs of the L3-L4, L4-L5, and L5-S1 levels were 8.2 ± 1.3 (mm, millimetre), 9.3 ± 1.4 (mm), and 8.9 ± 1.8 (mm), respectively. The mean DDs of the L3-L4, L4-L5, and L5-S1 levels were 31.3 ± 2.7 (mm), 30.8 ± 2.7 (mm), and 29.0 ± 2.6 (mm), respectively. The intrarater reliability of the MRI assessment for disc bulging and/or protrusion was 0.883, 0.833, and 0.883 at the L3-L4, L4-L5, and L5-S1 levels, respectively (Table 3). For the interrater and intrarater reliability of the disc morphology measurement, the ICCs for DH were 0.878 and 0.917 at the L3-L4 level, 0.899 and 0.916 at the L4-L5 level, and 0.943 and 0.948 at the L5-S1 level, respectively. The ICCs for DD at the L3-L4 level were 0.805 and 0.926, at the L4-L5 level were 0.939 and 0.963, and at the L5-S1 level were 0.858 and 0.991 for interrater and intrarater reliability, respectively (Table 4). High reliabilities for the morphology measurements in this study were found.

Table 5 shows the association between anthropometric factors, disc morphology and disc bulging and/or protrusion at the L3-L4, L4-L5, and L5-S1 levels. The resulting formula for logistic Model 1(Anthropometric model) at the L3-L4 level (disc bulging and/or protrusion) was 0.10 × Age + (−0.29) × Sex + (−0.01) × Body height + 0.04 × Body weight. Model 1 at the L4-L5 level was 0.01 × Age + 0.16 × Sex + 0.02 × Body height + 0.01 × Body weight. Model 1 at the L5-S1 level was 0.04 × Age + (−0.04) × Sex + 0.01 × Body height + 0.02 × Body weight. In Model 1 at the L3-L4, L4-L5 and L5-S1 levels, greater age and higher body weight were all significantly associated with disc bulging and/or protrusion. The resulting formula for logistic Model 2 (Disc morphology and anthropometric model) at the L3-L4 level was 0.08 × Age + (−0.10) × Sex + (−0.05) × Body height + 0.03 × Body weight + (−0.19) × Disc height + (0.37) × Disc depth. Model 2 at the L4-L5 level was 0.06 × Age + (−0.007) × Sex + (−0.05) × Body height + 0.03 × Body weight + (−0.18) × Disc height + 0.30 × Disc depth. Model 2 at the L5-S1 level was 0.04 × Age + 0.15 × Sex + (−0.005) × Body height + 0.02 × Body weight + (−0.12) × Disc height + 0.19 × Disc depth. In Model 2 (Disc morphology and anthropometric model) at the L3-L4, L4-L5 and L5-S1 levels, greater age, higher body weight, reduced disc height and increased disc depth were all significantly associated with disc bulging and/or protrusion. Body height was negatively associated with disc bulging and/or protrusion only at the L3-L4 and L4-L5 levels. Among the anthropometric variables, greater age and higher body weight were significantly associated with disc bulging and/or protrusion at the L3-L4, L4-L5 and L5-S1 levels, by both Model 1 and Model 2. Body height was negatively associated with disc bulging and/or protrusion at the L3-L4 and L4-L5 levels but not at the L5-S1 level. Sex was not associated with disc bulging and/or protrusion. Regarding the disc morphology variables, reduced disc height and increased disc depth were both significantly associated with disc bulging and/or protrusion at the L3-L4, L4-L5 and L5-S1 levels with both Model 1 and Model 2. The R square value of the Model 2 (Disc morphology and anthropometric model) were also greater than Model 1 (Anthropometric model) in all three disc levels. Those above results indicated that Model 2 (Disc morphology and anthropometric model) had a better capacity to predict disc bulging and/or protrusion compared with Model 1 using only anthropometric factors.

The ability of Model 1 and Model 2 to discriminate between participants with and without disc bulging and/or protrusion at the L3-L4, L4-L5 and L5-S1 disc levels are shown in Figure 2, Figure 3 and Figure 4, respectively. Compared with an AUC = 0.5, AUC statistics at the L3-L4 disc level were significantly different from 0.5 in both Model 1 (AUC = 0.77 [95% confidence interval (CI): = 0.73–0.81], *p* = 0.0001) and Model 2 (AUC = 0.81 [95% CI = 0.77–0.85], *p* = 0.0001). The AUC statistics were significantly better for Model 2 (Disc morphology and anthropometric model) than for Model 1 (Anthropometric model) (*p* < 0.05) (Figure 2). Regarding bulging/protrusion at the L4-L5 disc level, the AUC statistics were significantly different from 0.5 in both Model 1 (AUC = 0.74 [95% CI = 0.70–0.78], *p* = 0.0001) and Model 2 (AUC = 0.77 [95% CI = 0.73–0.81], *p* = 0.0001). The AUC statistic was significantly better for Model 2 (Disc morphology and anthropometric model) than for Model 1 (Anthropometric model) (*p* < 0.05) (Figure 3). Regarding bulging/protrusion at the L5-S1 disc level, the AUC statistics were significantly different from 0.5 in both Model 1 (AUC = 0.65 [95% CI = 0.61–0.70], *p* = 0.0001) and Model 2 (AUC = 0.67 [95% CI = 0.63–0.72], *p* = 0.0001). The AUC statistic for disc bulging and/or protrusion at the L5-S1 disc level was not significantly different from 0.5 for Model 1 or Model 2 (*p* > 0.05) (Figure 4). In summary, the AUC statistics were significantly better for Model 2 (Disc morphology and anthropometric model) than for Model 1 (Anthropometric model) at the L3-L4 and L4-L5 disc levels, indicating a better capability of determining the presence of disc bulging and/or protrusion by adding disc morphology factors to anthropometric factors (i.e., Model 2) than by using anthropometric factors alone. The sensitivity and specificity of each model represented positive results (Table 6). The positive predictive value was relatively low compared to the negative predictive value of each model. However, the positive predictive values in Model 2 (Disc morphology and anthropometric model) were better than in Model 1 (Anthropometric model) in all three disc levels.

## 4. Discussion

In this paper, in addition to anthropometric factors, disc morphology was found to be useful in predicting disc bulging and/or protrusion at the L3-L4 and L4-L5 levels but not at the L5-S1 level. Prediction models utilizing both anthropometric factors and disc morphology factors had a better capacity to predict disc bulging and/or protrusion compared with models using only anthropometric factors.

In the present study, our findings showed that a reduction in DH was associated with disc bulging and/or protrusion at the L3-L4, L4-L5 and L5-S1 levels, which was consistent with other studies. Brinckmann and Grootenboer found a disc height reduction and an increase in disc bulge in proportion to the amount of disc tissue removed [8]. In another study, the authors found that fracture and discectomy result in an increase in radial disc bulge and a decrease in disc height [9]. These studies revealed that there was a relationship between DH and disc bulge. Tibrewal also found that patients with disc herniation had reduced DHs compared with normal patients; unfortunately, the differences did not reach statistical significance [10]. The reason might be because of the smaller sample size in this study and the greater anatomic variation at the L5-S1 disc level. It has long been the clinical experience that patients with disc bulging/protrusion have disc space narrowing [10], and our study quantified this experience. In addition to DH, we observed that DD was closely related to disc bulging and/or protrusion at the L3-L4, L4-L5 and L5-S1 levels. It seems that not only DH but also DD was associated with disc bulging and/or protrusion. According to Natarajan’s study, changes in disc volume or disc area might be more related to disc bulging than a decrease in DH [26]. Therefore, we ought to take both DH and DD into account when considering disc bulging/protrusion.

Regarding anthropometric factors, the results showed that older age was associated with disc bulging and/or protrusion. A number of studies have indicated that ageing is an important risk factor for disc bulging and/or protrusion. Videman indicated that LDD, including disc dehydration and bulge and disc height narrowing, shows an increasing prevalence with age [16]. Another study reported that increasing age correlated with a higher prevalence of disc bulge [18]. Twomey showed that intervertebral discs become more convex in old age [12]. In addition, greater body weight was associated with disc bulging and/or protrusion, which was consistent with Hrubec’s results. He reported that body height and body weight were positively associated with the risk of disc herniation diagnosed in a United States Army hospital [14]. The literature focused on age and body weight as they relate to bulging/protrusion appears to be generally compatible with the results obtained in this study. However, body height was negatively associated with disc bulging and/or protrusion at the L3-L4 and L4-L5 levels in this study. Some studies have indicated that tallness is a factor associated with increased risk of herniation [14,15], but Kelsey’s studies failed to support such a relationship [27,28], as did our study. In a study on disc herniation, men with a height of 180 cm or more showed a relative risk of 2.3 and women with a height of 170 cm or more showed a relative risk of 3.7, compared with those who were more than 10 cm shorter. The author reported that body height may be an important contributor to the herniation of lumbar intervertebral discs [15]. The reason that our results did not show this relationship might be explained by the fact that the average body height (162.8 ± 7.9 cm) of our participants was not as high as that of the previous study.

One thing that should be considered is the diurnal variation of the intervertebral disc in the disc height measurement. The diurnal variation in disc height was reported to be similar in the lower three lumbar discs [29]. The MRI assessments in this study were taken between 3 and 6 h after rising, and the diurnal loss was considered to be similar among the participants. Therefore, we assumed that the diurnal change was unlikely to have influenced the measurement of disc height.

At the L5-S1 disc level, the ability to predict disc bulging and/or protrusion of Model 2 (Disc morphology and anthropometric model) was not significantly better than that of Model 1 (Anthropometric model). The results suggest that disc morphology might not have a sufficient effect on bulging/protrusion at the L5-S1 level. The reason could be due to the wide individual variation in the L5-S1 disc. The lumbar disc height generally increased towards the lower lumbar level, except for L5-S1. Accordingly, narrowing of disc height is usually determined clinically on plain radiographs in comparison with the adjacent disc height, particularly that one level above. However, narrowing of the L5-S1 disc height was difficult to judge on plain radiographs [30]. For these reasons, disc morphology might not be useful for predicting disc bulging and/or protrusion at the L5-S1 level.

In this study, we included disc bulging and protrusion as outcomes for several reasons. From the literature review, not only was the relationship between disc height and disc bulging found, but so was a relationship between disc height and disc protrusion. In the clinic, cases that were diagnosed as disc bulging had the potential to become disc protrusion; even cases that were diagnosed as disc protrusion sometimes had the opportunity to reverse to disc bulging. In addition, some cases might be diagnosed as disc bulging and disc protrusion at the same time by different radiologists. In some difficult cases, it is difficult to distinguish between disc bulging and/or protrusion. In addition, there were only four subjects with extruding discs and one subject with a sequestrated disc among the original 553 participants. After excluding a total of 101 participants with degenerative spondylolisthesis, spondylolytic spondylolisthesis and a previous lumbar spine operation, none of the participants were found to have extrusion or sequestration among 452 participants. For those reasons, we decided to extend the range of the outcome by combining cases with disc bulging and cases with disc protrusion for better prediction.

There were several advantages in this study. To the best of our knowledge, this study is one of the few studies to research both anthropometry and disc morphology factors in relation to disc herniation and to further develop a disc bulging and/or protrusion predictive model by using disc height, disc depth, and anthropometry factors. We also investigated multiple disc levels, i.e., L3-L4, L4-L5 and L5-S1, compared to other studies that investigated a single disc level. Large sample sizes of MRI scans were obtained, and standard images were used to characterize the presence of disc bulging and/or protrusion in an Asian population. In addition, the evaluation of disc bulging and/or protrusion and measurements of disc morphology were taken without knowledge of other demographic, clinical, and imaging information. Hence, the errors in classification could likely have been prevented. It is valuable to provide DH and DD quantitatively so that the data can be used to predict disc bulging and/or protrusion.

There are several limitations in this study that need to be considered when interpreting the results. One of the limitations of this study is that the effect of ageing on DH and DD may cause overestimation by the prediction model. Several studies have reported DH or DD being related to age [12,16,25,26,31,32]. Natarajan found a decline in DH after the fifth decade of life [26]. Amonookuofi showed that disc height and diameter vary significantly in different age groups [25]. The sizes of discs increase as a person ages [25]. In another study, the maximum DH was greater in older individuals (50–60 years) than in younger individuals (20–30 years) [32]. This was presumed to be a result of microfracturing of the endplate during adult life, which leads to a more concave form of intervertebral disc [32]. However, Koeller had different observations, i.e., that average DH is almost independent of age [33]. Therefore, we assessed the correlation of age, DH and DD. The correlation between age and DH was r (correlation coefficient) = −0.025, *p* > 0.05 at the L3-L4 level; r = −0.063, *p* > 0.05 at the L4-L5 level; and r = 0.116, *p* = 0.01 at the L5-S1 level. The correlation between age and DD was r = 0.151, *p* = 0.001 at the L3-L4 level; r = 0.180, *p* = 0.0001 at the L4-L5 level; and r = 0.121, *p* = 0.009 at the L5-S1 level. The results showed that the correlation between age and disc morphology was low in our study. Furthermore, we adjusted age to minimize any confounding that might occur. After the adjustment, we assumed that any overestimation by the predictive model would not be serious. The cause of disc herniation is multifactorial, including lifting load, award posture, genetic factors, as well as anthropometric factors. Battie’s study indicated that [13,18] with the addition of familial aggregation to the model, the explained variability in the discs increased from 9% to 43%. Thus, disc degeneration can be explained significantly by genetic effect. This might be one of the reasons for which the explanatory power of the predictive models in our study were small, as well as the predictive positive value. Furthermore, there could be selection bias due to having relatively healthy participants in the study. The participants had to be mobile enough to visit the NTUH to undergo the MRI assessment. In other words, participants with more severe symptoms and pain might not be included. There might be a healthy worker effect in this study. In addition, since the participants in this study were market workers and walk-in patients to the Internal Medicine clinic for treatment of a common cough, low back pain was not the main complaint for treatment. We did not obtain a medical diagnosis regarding back pain conditions or treatments. Therefore, without precise further medical examination and diagnoses, we could not determine whether disc bulging and/or protruding were the main cause of low back pain in those participants with objective back pain complaints (70.0%).

## 5. Conclusions

In conclusion, our data provide evidence for an association between disc morphology and disc bulging and/or protrusion at the L3-L4, L4-L5 and L5-S1 levels. Furthermore, a prediction model using both anthropometric factors and disc morphology factors to predict disc bulging and/or protrusion at the L3-L4 and L4-L5 levels was developed. This prediction model could be used on plain films to facilitate clinical diagnosis and increase medical resource utilization.

## Figures and Tables

**Figure 1 ijerph-18-02521-f001:**
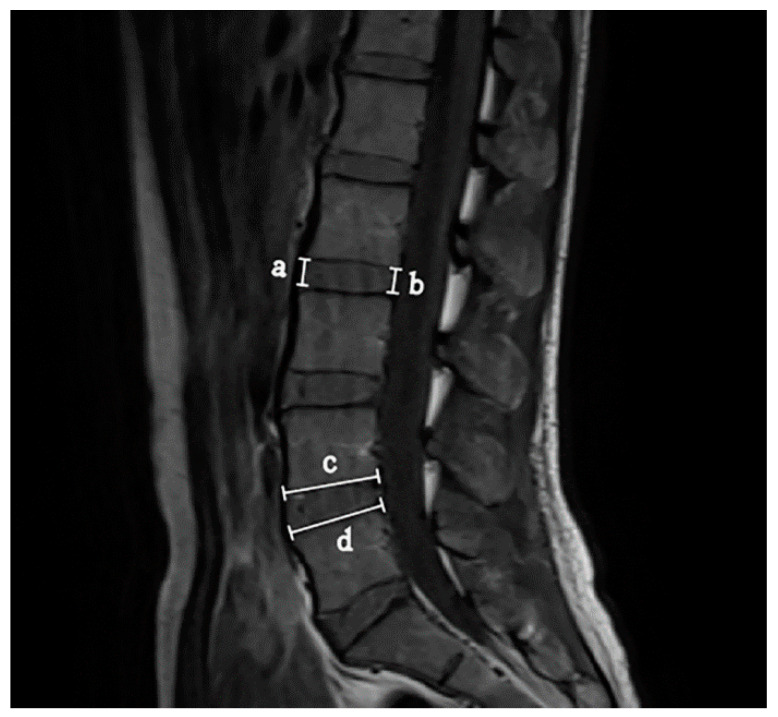
Illustration of measurement of intervertebral disc height and disc depth. a: Anterior disc height (ADH), b: Posterior disc height (PDH), Disc height (DH) = 1/2 (ADH + PDH) = 1/2 (a + b). c: Superior disc depth (SDD), d: Inferior disc depth (IDD), Disc depth (DD) = 1/2 (SDD + IDD) = 1/2 (c + d).

**Figure 2 ijerph-18-02521-f002:**
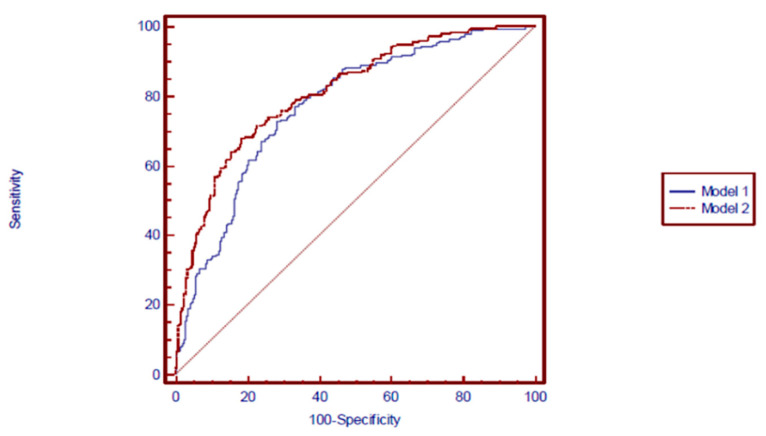
Receiver-operating characteristic curves for the prediction of L3-L4 disc bulging and/or protrusion by model 1 and model 2. Model 1 (Anthropometric model): Area under the curve (AUC) (95% CI) = 0.77 (0.73–0.81). *p* = 0.0001. Model 2 (Disc morphology and anthropometric model): AUC (95% CI) = 0.81 (0.77–0.85). *p* = 0.0001 *p*-value for comparison of AUCs < 0.05.

**Figure 3 ijerph-18-02521-f003:**
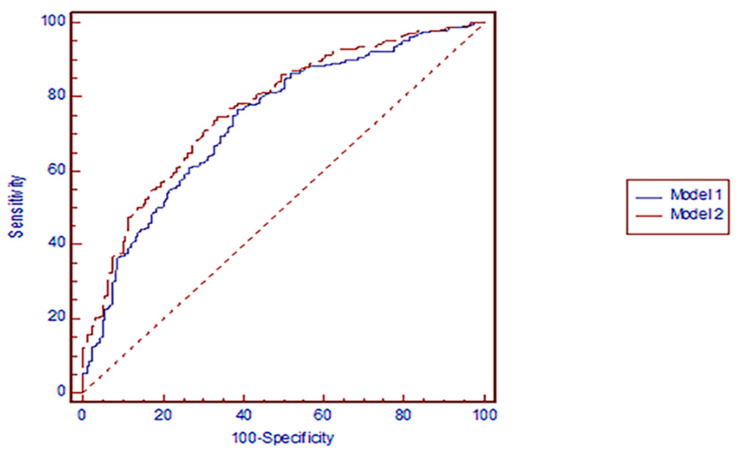
Receiver-operating characteristic curves for the prediction of L4-L5 disc. bulging and/or protrusion by model 1 and model 2. Model 1 (Anthropometric model): AUC (95% CI) = 0.74 (0.70–0.78). *p* = 0.0001, Model 2 (Disc morphology and anthropometric model): AUC (95% CI) = 0.77 (0.73–0.81). *p*= 0.0001, *p*-value for comparison of AUCs < 0.05.

**Figure 4 ijerph-18-02521-f004:**
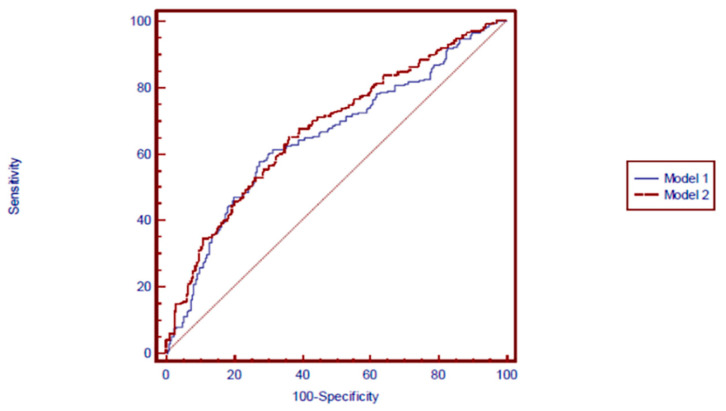
Receiver-operating characteristic curves for the prediction of L5-S1 disc bulging and/or protrusion by model 1 and model 2. Model 1 (Anthropometric model): AUC (95% CI) = 0.65 (0.61–0.70). *p* = 0.0001, Model 2 (Disc morphology and anthropometric model): AUC (95% CI) = 0.67 (0.63–0.72). *p* = 0.0001, *p*-value for comparison of AUCs > 0.05.

**Table 1 ijerph-18-02521-t001:** Demographic characteristics of the study participants.

Variables	Mean (Range)	*N* (%)
Age (years)	49.3 (21–65)	
<40		88 (19.5)
40~<50		110 (24.3)
50~<60		188 (41.6)
≥60		66 (14.6)
Sex		
Male/Female		210 (46.5)/242(53.5)
Body height (cm)	162.8 (140.0–185.0)	
Body weight (kg)	65.5 (39.5–113.0)	
BMI (kg/m^2^)	24.6 (16.6–39.0)	
<24		191 (42.3)
24~<27		159 (35.2)
≥27		102 (22.6)
Low back pain (within 12 months)		
Yes		313 (70.0)
missing		5 (1.1)
Work tenure (years)	22.5(0.63–50)	
<15		120 (26.5)
15~<30		177 (39.1)
≥30		155 (34.4)
Exercise (Yes)		294 (65.0)
missing		4 (0.9)
Smoking (Yes)		103 (22.8)
Education Level		
Junior high and below		142 (31.4)
Senior high school		216 (47.8)
College or above		89 (19.7)
Missing		5 (1.1)

**Table 2 ijerph-18-02521-t002:** The disc morphology factors of the L3-L4, L4-L5, and L5-S1 levels.

Disc Level	Disc Height (mm) Mean (Range)	Disc Depth (mm) Mean (Range)
L3-L4	8.2 (4.8–11.6)	31.3 (20.9–40.8)
L4-L5	9.3 (4.1–13.1)	30.8 (21.9–39.8)
L5-S1	8.9 (3.4–13.3)	29.0 (21.1–37.0)

**Table 3 ijerph-18-02521-t003:** Intra-rater reliability of disc bulging and/or protrusion in MRI by percentage agreement.

Variable	Disc Level	Intra-Rater Reliability
Disc bulging and/or protrusion	L3-L4	0.883
	L4-L5	0.833
	L5-S1	0.883

**Table 4 ijerph-18-02521-t004:** Intra-rater and inter-rater reliability of disc height and disc depth measurement by interclass correlation coefficients (ICC).

Variable	Disc Level	Inter-Rater Reliability	Intra-Rater Reliability
Disc height	L3-L4	0.878	0.917
	L4-L5	0.899	0.916
	L5-S1	0.943	0.948
Disc depth	L3-L4	0.805	0.926
	L4-L5	0.939	0.963
	L5-S1	0.858	0.991

**Table 5 ijerph-18-02521-t005:** The association between anthropometric factors and disc bulging and/or protrusion, and anthropometric factors with disc morphology and disc bulging and/or protrusion at the L3-L4, L4-L5, and L5-S1 levels by logistic regression.

L3-L4 Disc Bulging and/or Protrusion					L4-L5 Disc Bulging and/or Protrusion				L5-S1 Disc Bulging and/or Protrusion			
	β	Standardizedβ	*P*	R^2^	β	Standardized β	*P*	R^2^	β	Standardized β	*P*	R^2^
Model 1 (Anthropometric model)				0.288				0.213				0.088
Age	0.10	0.0013	<0.0001		0.01	0.0001	<0.0001		0.04	0.0004	<0.0001	
Sex	−0.29	−0.0476	0.08		0.16	0.0258	0.13		−0.04	−0.0060	0.8	
Body height	−0.01	−0.0002	0.58		0.02	0.0004	0.55		0.01	0.0002	0.56	
Body weight	0.04	0.0005	0.0025		0.01	0.0001	0.003		0.02	0.0002	0.03	
Model 2 (Disc morphology and anthropometric model)				0.371								
Age	0.08	0.0010	<0.0001		0.06	0.0007	<0.0001		0.04	0.0004	0.0008	
Sex	−0.10	−0.0179	0.59		−0.007	−0.0012	0.97		0.15	0.0246	0.36	
Body height	−0.05	−0.0012	0.02		−0.05	−0.0012	0.04		−0.005	−0.0001	0.82	
Body weight	0.03	0.0004	0.02		0.03	0.0004	0.01		0.02	0.0002	0.036	
Disc height	−0.19	−0.0179	0.038		−0.18	−0.0151	0.03		−0.12	−0.0070	0.046	
Disc depth	0.37	0.0248	<0.0001		0.30	0.0192	<0.0001		0.19	0.0106	0.0009	

**Table 6 ijerph-18-02521-t006:** The sensitivity, specificity, positive and negative predictive value of the predictive model.

Disc Level	Predictive Model	Sensitivity (%)	Specificity (%)	Criterion	Positive Predictive Value (%)	Negative Predictive Value (%)
L34	Model 1 (Anthropometric model)	72.9	71.9	>0.4568	1.13	99.83
	Model 2 (Disc morphology and anthropometric model)	67.8	81.4	>0.5155	1.59	99.83
L45	Model 1 (Anthropometric model)	76.6	61.5	>0.5656	1.19	99.77
	Model 2 (Disc morphology and anthropometric model)	74.7	66.5	>0.5673	1.34	99.77
L5S1	Model 1 (Anthropometric model)	57.9	72.5	>0.4625	0.91	99.75
	Model 2 (Disc morphology and anthropometric model)	65.0	63.5	>0.4230	0.77	99.76

## Data Availability

The dataset used for analysis during the current study are available from the corresponding author on reasonable request.

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
