# Peer review of "Prediction of Lumbar Disc Bulging and Protrusion by Anthropometric Factors and Disc Morphology"

_ijerph, 2021, doi:10.3390/ijerph18052521_

Round 1
Reviewer 1 Report
Introduction
The final part of the introduction should be modified. It is not convenient to reference bibliographically the text that appears in the objective.
In the final part of the introduction, the authors should introduce a paragraph explaining which variables have been associated with LDD and disc herniation. Then, add the objectives of the study.
Material and Methods
The material and methods should be restructured, separating how the sample was accessed, data collection and the ethical section. Within the data collection section, the current 2.2, 2.3, 2.4 can be placed as subsections 2.2, 2.3, 2.4.
The sample size section should be placed in the subjects section.
Why is it not specified how the anthropometric variables were collected?
Results
Table 1 shows not only asbolute and relative frequencies, but also means and standard deviations. The table is presented in a complicated way.
Sex and gender are used indifferently, and they should not be.
Table 5 should show the standardised beta values for each variable and the adjusted R-squared for each model.
Why are variables above 0.25 kept in the models?
What is the sensitivity, specificity, positive and negative predictive value of each model?
The three ROC curves should be in one figure.
I cannot assess the discussion and conclusion without some of the missing data in the results.
Author Response
Introduction
Point 1. The final part of the introduction should be modified. It is not convenient to reference bibliographically the text that appears in the objective.
Response 1: Thanks for your advice. We had modified the introduction part at page 4-5.
Point 2. In the final part of the introduction, the authors should introduce a paragraph explaining which variables have been associated with LDD and disc herniation. Then, add the objectives of the study.
Response 2: Thanks for your advice. We added a paragraph before the final part of the introduction, page 4-5, as following “In Videman’s study, disc herniation was associated with larger axial disc area [13]. Regarding anthropometric factors, Hrubec reported that body height and body weight were positively associated with the risk of disc herniation [14]. Other studies also indicated that tallness is a factor associated with increased risk of herniation [14,15]. In addition, a number of studies have indicated that ageing is an important risk factor for disc bulging and/or protrusion [12,14,16-18]. From the literature review, disc morphology such as disc height and disc depth, and the anthropometric factors like age, sex, body height, and body weight had the relationship been LDD and disc herniation.”
Material and Methods
Point 3. The material and methods should be restructured, separating how the sample was accessed, data collection and the ethical section. Within the data collection section, the current 2.2, 2.3, 2.4 can be placed as subsections 2.2, 2.3, 2.4.
Response 3: Thanks for your suggestion. The material and methods was restructured as the following: the ethical section was moved to the first paragraph of the 2.1. Subjects recruitment subsection. The subsection were as following, page 6-11”
2.1. Subjects recruitment
2.2 data collection
2.2.1. Magnetic resonance imaging (MRI) equipment and protocol
2.2.2. Definition of disc bulging, disc protrusion, degenerative spondylolisthesis
and spondylolytic spondylolisthesis
2.2.3. Disc morphology measurement
2.3. Statistical analysis
Point 4. The sample size section should be placed in the subjects section.
Response 4: Edited as recommend, page 6, 7.
Point 5. Why is it not specified how the anthropometric variables were collected?
Response 5: Thanks for your reminding. We added it in 2.2 data collection subsection, page 7, as the following” The content of the questionnaire each participant completed consisted of questions regarding demographic information including age, sex, body height, and body weight,………………”
Results
Point 6. Table 1 shows not only asbolute and relative frequencies, but also means and standard deviations. The table is presented in a complicated way.
Response 6: Thank you for your comment. We had added one column of means and range in Table 1 (page 15) for presenting in a clear way.
Point 7. Sex and gender are used indifferently, and they should not be.
Response 7: Thank you for reminding us. We edited the term ”gender” to “Sex” for the consistency, in Table 1, page 15.
Point 8. Table 5 should show the standardised beta values for each variable and the adjusted R-squared for each model.
Response 8: Thank you for your advice. Table 5 (page19-20) showed the association between the risk factors and the disc bulge buy using logistic regression. However, the standardized beta values for each variable and the adjusted R-squared in logistic regression model were not provided by the statistical software we used. However, we have obtained “Nagelkerke R square” from SPSS. In addition, we tried to calculate the standardized beta value by using the method (1.2), as the reference shown below. We hope the above results will be satisfactory.
(Hong, CS and Ryu,HS. Information Theoretic Standardized Logistic Regression Coefficients with Various Coefficients of Determination. The Korean Communications in Statistics.Vol(13), pp. 49-60, 2006)
We also added the sentences in Results section, page 18, as following ”The R square value of the Model 2 (Disc morphology and anthropometric model) were also greater than Model 1(Anthropometric model) in all three disc levels. Those above results indicated that Model 2 (Disc morphology and anthropometric model) had a better capacity to predict disc bulging and/or protrusion compared with Model 1 using only anthropometric factors.”
Point 9. Why are variables above 0.25 kept in the models?
Response 9: Thanks for the comment. To clarify this issue, we added the sentences in the 2.3. Statistical analysis subsection, page12, as following: ” In addition, variables that were based on the findings from past research were also selected for use in the model, such as body height was reported that associated with disc herniation [14].
Point 10. What is the sensitivity, specificity, positive and negative predictive value of each model?
Response 10: Thanks for your valuable advice. We had added the sensitivity, specificity, positive and negative predictive value of each model in Table 6, page 21.
We also added the paragraph in the final part of the Result section, page 23, as the following:” The sensitivity and specificity of each model represented positive results (Table 6). The positive predictive value was relatively low than negative predictive value of each model. However, the positive predictive values in Model 2 (Disc morphology and anthropometric model) were better than in Model 1(Anthropometric model) in all three disc levels.”
And added the paragraph in the limitation of the Discussion section, page 31-32, as following:” The cause of disc herniation is multifactorial, including lifting load, award posture, genetic factors, as well as anthropometric factors. Battie’study indicated that [13,18] with the addition of familial aggregation to the model, the explained variability in the discs increased from 9% to 43%. Thus, disc degeneration can be explained significantly by genetic effect. This might be one of the reason that the explanatory power of the predictive models in our study were small, as well as the predictive positive value.”
Point 11. The three ROC curves should be in one figure.
Response 11: Thank you for your advice. We agree with this suggestion, however given there were 3 dependent variables (i.e. L34,L45 and L5S1), we feel it would be clearer for the readers to see the fit of each model by presenting 3 separate ROC graphs.
Point 12. I cannot assess the discussion and conclusion without some of the missing data in the results.
Response 12: We hope the above results will be satisfactory.
We would like to thank you for your efforts in review of this manuscript. In summary, we are grateful for how your feedbacks and comments have benefited and advanced our approach to the study. We hope that the revised manuscript is more suitable for publication in IJERPH. Your kind attention and consideration on this manuscript will be most appreciated.
Sincerely Yours,
Yue-Liang Leon Guo, MD, MPH, PHD
Professor and Director, Environmental and Occupational Medicine
National Taiwan University and NTU Hospital
Taipei 100
TAIWAN
Reviewer 2 Report
The authors present novel data on an “old” problem how disc bulging and/or protrusion correlates with disc height decrease. They created two distinct models: model 1 was based on “anthropometric” factors and linking this to surgery outcome. Model 2 contained more disc morphology.
The English are of a good quality, no improvements needed as far I can judge this.
The conclusions are fine.
Specific comments.
The material and methods are described in enough detail, such that the study could be repeated.
The MRI procedures seem to be given in enough detail and can be reproduced with the provided information.
Discussion
The discussion reads fine.
The bibliography looks in the right format. Possibly the DOI numbers need to be added for MDPI? It also seems that the most relevant citations were added.
Figure legends: I had the impression that the text could be a bit extended such that the legend is a bit more independent from the main text. E.G., …prediction of L5-S1 INTERVERTEBRAL disc bulging by model 1 and model 2. Maybe the designations of the two models could also a bit more self-explaining rather than just to give it a number. Like “Model 1= anthropometric model” and “model 2 = disc morphology model”. Maybe like this the figure legends would also stand on their own and could be easier understood without reading the main text.
Author Response
Point 1: The bibliography looks in the right format. Possibly the DOI numbers need to be added for MDPI? It also seems that the most relevant citations were added.
Response 1: Thank you for your suggestion. We had added the DOI numbers to the reference articles except several ones that they did not provided.
Point 2: Figure legends: I had the impression that the text could be a bit extended such that the legend is a bit more independent from the main text. E.G., …prediction of L5-S1 INTERVERTEBRAL disc bulging by model 1 and model 2. Maybe the designations of the two models could also a bit more self-explaining rather than just to give it a number. Like “Model 1= anthropometric model” and “model 2 = disc morphology model”. Maybe like this the figure legends would also stand on their own and could be easier understood without reading the main text.
Response 2: Thank you for reminding us to clarify this issue. We had revised it as “Model 1(Anthropometric model)” and “Model 2 (Disc morphology and anthropometric model)” to Figure legends, as well as in main text.
We would like to thank you for your efforts in review of this manuscript. In summary, we are grateful for how your feedbacks and comments have benefited and advanced our approach to the study. We hope that the revised manuscript is more suitable for publication in IJERPH. Your kind attention and consideration on this manuscript will be most appreciated.
Sincerely Yours,
Yue-Liang Leon Guo, MD, MPH, PHD
Professor and Director, Environmental and Occupational Medicine
National Taiwan University and NTU Hospital
Taipei 100
TAIWAN
Round 2
Reviewer 1 Report
Congratulations on your work.